# Early Warning of Gas Concentration in Coal Mines Production Based on Probability Density Machine

**DOI:** 10.3390/s21175730

**Published:** 2021-08-25

**Authors:** Yadong Cai, Shiqi Wu, Ming Zhou, Shang Gao, Hualong Yu

**Affiliations:** School of Computer, Jiangsu University of Science and Technology, Zhenjiang 212100, China; 199070029@stu.just.edu.cn (Y.C.); shiqi_wu@stu.just.edu.cn (S.W.); 199070030@stu.just.edu.cn (M.Z.); gao_shang@just.edu.cn (S.G.)

**Keywords:** gas concentration, coal mines, early warning, class imbalance learning, probability density estimation

## Abstract

Gas explosion has always been an important factor restricting coal mine production safety. The application of machine learning techniques in coal mine gas concentration prediction and early warning can effectively prevent gas explosion accidents. Nearly all traditional prediction models use a regression technique to predict gas concentration. Considering there exist very few instances of high gas concentration, the instance distribution of gas concentration would be extremely imbalanced. Therefore, such regression models generally perform poorly in predicting high gas concentration instances. In this study, we consider early warning of gas concentration as a binary-class problem, and divide gas concentration data into warning class and non-warning class according to the concentration threshold. We proposed the probability density machine (PDM) algorithm with excellent adaptability to imbalanced data distribution. In this study, we use the original gas concentration data collected from several monitoring points in a coal mine in Datong city, Shanxi Province, China, to train the PDM model and to compare the model with several class imbalance learning algorithms. The results show that the PDM algorithm is superior to the traditional and state-of-the-art class imbalance learning algorithms, and can produce more accurate early warning results for gas explosion.

## 1. Introduction

Coal resource is an important basic energy source in China. From 2015 to 2019, the average annual coal production in China accounted for about 68% of the total energy production [1]. However, coal mine safety accidents have caused serious economic losses and also seriously endangered the lives of miners. Gas explosions are an important cause of coal mine safety accidents, which may cause many miners to die or can destroy the whole coal mine [2]. According to statistics, from 2007 to 2017, gas explosions accounted for 50% of coal mine safety accidents in China. From 2004 to 2015, 10,298 persons died from gas explosion accidents, accounting for 29.7% of deaths in various coal mine accidents [3,4]. Therefore, the prevention of gas explosion accidents should be the top priority of coal mine safety accident prevention in China. How to reduce such hazards and to achieve safe coal mining is a major problem. It is of great significance to improve the ability of gas disaster prediction by strengthening the study of gas disaster prediction and early warning technology.

In recent years, researchers have found that the gas concentration generally rises abnormally before incidents of coal mine gas explosions [5]. Therefore, in order to predict gas concentration accurately and prevent gas accidents effectively, researchers have proposed some gas prediction methods. Models of gas concentration or gas outburst forecasts are largely based on BP neural networks [6,7,8], LSTM neural networks [9], the SVR algorithm [10,11], the ELM algorithm [12], the Gaussian process regression algorithm [13], and some other mathematical or statistical methods [14,15,16]. These methods always use the time-series data collected by the gas sensors to establish the regression prediction model of gas concentration. However, we note that the number of the actual gas data collected by the sensors exceeding the warning threshold is scarce, so the traditional warning models cannot learn the rule for when the gas concentration rises abnormally, thus it is difficult to achieve the effect of warning in advance. 

In this study, we consider gas early warning prediction as a binary classification issue. Specifically, the sensing data are divided into one of two classes, early warning class and non-early warning class, based on a pre-defined early warning threshold. It is reasonable that regarding gas early warning prediction as a binary classification issue, owing to in practical coal mining production, it is not necessary to accurately predict gas concentration over a period of future time, but should judge whether it has a high risk to impact safety production or not. Several previous studies support this system of construction, e.g., Ruta and Chen [17] constructed a methane concentration warning model by combining multiple classification models with optimization approach to provide a relatively accurate warning for the future 3 min’ methane emission; Zhang et al. [18] developed a gas outburst early warning system by adopting an entropy-weight Bayes inference model. These models seem to be effective, however, they all ignore an important data character, i.e., these kind of data are always extremely imbalanced.

To deal with class imbalanced data, some different solutions have been proposed, and they can be roughly divided into two groups: data level and algorithm level. Data level generally adopts resampling strategies to increase instances belonging to the minority class, or to decrease instances from the majority class, and further re-balances the data distribution [19,20,21]. Several popular resampling algorithms include random undersampling (RUS) [22], random oversampling (ROS) [22] and the synthetic minority oversampling technique (SMOTE) [23] etc. ROS tends to be overfitting, RUS is apt to lose some key information related with classification, while SMOTE is inclined to propagate noises. In recent years, some advanced sampling algorithms have also been proposed to address the problems mentioned above. For example, Xie et al. [24] proposed a GL algorithm which first takes advantage of the mixture-Gaussian model to estimate the distribution of minority instances, and then oversamples minority class based on the estimated results. As for the algorithm-level strategy, it mainly includes cost-sensitive learning [25,26] and the threshold moving technique [27,28]. Cost-sensitive learning designates different penalties for training errors belonging to different classes, further balancing the training errors of different classes. The threshold moving technique firstly trains a classification model, and then moves classification hyperplane towards majority class to repair the bias.

To promote the modeling quality of the early warning model on skewed gas concentration data, in this study we proposed a new class imbalance learning solution called the probability density machine (PDM), which adopts a KNN-PDE K nearest neighbors probability density estimation (KNN-PDE)-alike algorithm [29] to approximately estimate the probability density of each instance, and then directly compares the probability density of an instance on each class to decide which category that instance belongs to. The PDM algorithm has a good adaptability to the skewed data distribution. The PDM algorithm can directly achieve a good warning effect on gas concentration monitoring data, even without resampling any instances. The effectiveness and superiority of the PDM is verified on six gas concentration monitoring data sets collected from a coal mine in China.

The rest of this study is organized as follows. Section 2 reviews the corresponding methods related with the gas concentration prediction. Section 3 describes the procedure of data collection, data preprocessing and instance generation. In Section 4, we firstly analyze why imbalanced data distribution always hurts the performance of predictive models in context of Gaussian Naive Bayes (GNB) as our proposed PDM algorithm inherits from GNB [30,31], and then based on the analysis, we describe the proposed KNN-PDE-alike and PDM algorithms in detail. Section 5 analyzes and discusses the experimental results. Finally, Section 6 concludes this paper. 

## 2. Related Works

In recent years, researchers focused more and more attentions on the gas concentration and gas outburst prediction in coal mine safe production, and have presented some solutions.

Zhang et al. [6] noted that gas disasters are related to many factors, including crustal stress, coal structural performance, geological structure, gas content, etc. They combined the GM (1, 1) grey prediction model and BP neural network to predict the gas emission value, and acquired an improved result in comparison with only adopting any one single model. A similar method was proposed by Wang et al. [7], which also adopted an artificial neural network to handle various factors and various nonlinear relationships in geological conditions, and acquired an excellent prediction result about gas outburst. To avoid the BP neural network falling into local minimization, Wu et al. [8] combined genetic algorithm (GA) and the simulated annealing algorithm (SA) as a new genetic algorithm to improve the generalization of the neural network, and further improve prediction accuracy of gas outburst. 

Lyu et al. [9] first fused the gas information of multiple sensors inside the coal mine, and then used the LSTM model based on encoder-decoder to construct multivariant regression and predict the short-term gas concentration. Wu et al. [10] firstly used the t-distributed stochastic neighbor embedding (t-SNE) algorithm to perform non-linear dimensionality reduction in coal mine gas-related multi-dimensional monitoring data, then extracted the spatial feature data of the monitoring data, and finally used the support vector regression (SVR) algorithm to predict the top corner gas concentration. Meng et al. [11] also adopted SVR to predict mine gas emission rate and found it outperforms artificial neural networks. Wu et al. [12] decomposed the time series of gas concentration into many time-frequency components by using the wavelet analysis algorithm, and then constructed the prediction model of gas concentration by adopting the extreme learning machine (ELM). Dong et al. [13] combined the Bayesian network method, chaotic phase space reconstructive technology, and Gaussian process regression model to construct a gas concentration prediction model that can produce competitive prediction results.

The above-mentioned related studies considered the sophisticated geological factors which have a strong association with coal mine exploration, but a weak association with the process of coal mines production; or they regarded multiple different groups of sensing and monitoring data, and adopt multivariant regression to construct the prediction model. In this paper, we focus on a single variance, i.e., the gas concentration monitoring data acquired from a single sensor. We wish to analyze each single time-series and discover the regular temporal patterns from each time-series. In addition, we note in gas concentration monitoring sequence, the high concentration corresponds significantly less points, thus the data hold extremely imbalanced distribution. However, all existing models almost ignore this problem, which may be a great risk to subsequent modeling. It is also an important consideration in our study.

## 3. Data

### 3.1. Data Acquisition

We collected data by gas concentration sensors from 363 monitoring points in a coal mine in Datong city, Shanxi Province, China. The data are collected in the form of a time series during the period of 2019.10.22 00:00:00~2020.03.24 00:00:00, a total of 5 months. Figure 1 presents the original gas concentration variation trend of a monitoring point during these 5 months, where each point in the horizon-axis represents the average value of gas concentration in 10 min. In Figure 1, there is no clear law to describe the global gas concentration variation, but some local variation rules can be observed. In other words, the gas concentration of a time point can be only decided by its nearest short-term time series fragment.

### 3.2. Data Preprocessing

The actual operation of the gas data collector is affected by various factors such as the changes in the downhole environment, so that the actual monitoring data obtained are not a data sequence with uniform time intervals. Therefore, we first normalize each time series, taking 10 min as an interval, and adopting the average value of the received gas concentration readings existing in each 10 min as the representative of gas concentration for the corresponding 10 min.

Specifically, we note that if there were no readings within any one 10 min in the original sequence, then it would be represented as a null value of the corresponding position in the new normalized sequence. The statistical analysis of the original gas concentration data in coal mines shows that in the collected gas concentration data, the missing rate of a large number of monitoring points exceeds 20% during the period of 2019.10.22 00:00:00~2020.03.24 00:00:00, which might greatly affect the downstream modeling and analysis. Therefore, we only selected gas concentration data from six monitoring points with a missing rate of less than 20% in this study. These six monitoring points are respectively the return air gas face on surface 8301, the working face gas on surface 8301, the top corner gas on surface 8301, the top corner gas on surface 8222, the return air gas face on surface 8222, and the working face gas on surface 8222 (we call them D1, D2, D3, D4, D5 and D6 in brief). Figure 1 shows the variation trend of gas concentration in data set D1, and the variation trend of gas concentration in data sets D2–D6 are shown in Figure A1, Figure A2, Figure A3, Figure A4 and Figure A5, respectively. The statistical results of these data sets are shown in Table 1. 

In Table 1, it can be seen that these six data sets have 6.1~15.5% missing rate, which further destroys the downstream modeling procedure. To address this problem, we adopted a linear interpolation-based missing value imputation strategy. Assuming there exists *c* continuous missing values, the former non-missing value is *a*, and the latter non-missing value is *b*, then the difference is *d = b − a*, the step is e=d/c, thus based on the concept of interpolation, the imputed *c* missing values are a+1×e, a+2×e,…, a+c×e.

Figure 2a,b show the variation trend of gas concentration before and after imputing the missing data within a week. From Figure 2b, we can see that the gas concentration variation trend is more continuous and regular after data imputation, which verifies its rationality.

### 3.3. Instance Generation

Next, we need to consider how to take advantage of the imputed time series to generate instances, and to use them for training the classification model. As referred above, the variation in gas concentration is local, but not global, which means a future short-time gas concentration variation only relates with a recently happened fragment of gas concentration variation. Therefore, we adopt the slide-time-window strategy to generate instances (see Figure 3).

In Figure 3, we can see that in time series data, each slide time window corresponds with an instance, and when the window slides forward a step, it can acquire a new instance. Suppose a time series can be represented as $T=\{x_{1},x_{2},x_{3},\ldots ,x_{N-1},x_{N}\}$, the length of attributes, predictive values, and slide time window are respectively *l*, *t*, and *s*, where *s* = *l* + *t*, then it would generate *N* − *s* + 1 instances. Here, *l* denotes how long of recently monitored data can be used to predict future status, while *t* indicates how far of the future status can be predicted. Considering the high frequency of variation in gas concentration monitoring data, we designate *l* = 24 and *t* = 6 empirically in this paper. That is to say, we use the recent 4 h of experience to predict the future 1 h status.

Then the input matrix of the generated data set can be expressed as:(1)$$X=\left[\begin{array}{cccc} x_{1} & x_{2} & \ldots & x_{l}\\ x_{2} & x_{3} & \ldots & x_{l+1}\\ \vdots & \vdots & \ddots & \vdots \\ x_{N-l-t+1} & x_{N-n-t+2} & \ldots & x_{N-t+1} \end{array}\right]$$
and the output matrix represents as:(2)$$Y=\left[\begin{array}{c} x_{l+1}\\ x_{l+2}\\ \vdots \\ x_{N-t+1} \end{array}\begin{array}{c} \ldots \\ \ldots \\ \ddots \\ \ldots \end{array}\begin{array}{c} x_{l+t}\\ x_{l+t+1}\\ \vdots \\ x_{N} \end{array}\right]$$

We consider the early warning as a classification task in this study, thus the readings in the expected output Y should be divided into two categories according to a previously given early warning threshold of gas concentration. When the reading exceeds the threshold, the expected output is expressed as +1, indicating it belongs to the class of early warning, otherwise, the expected output is expressed as −1, indicating it belongs to the category of non-early warning. Then, the original output matrix Y is transformed to be the form below:(3)$$Y=\left[\begin{array}{c} -1\\ +1\\ \vdots \\ -1 \end{array}\begin{array}{c} \ldots \\ \ldots \\ \ddots \\ \ldots \end{array}\begin{array}{c} -1\\ -1\\ \vdots \\ +1 \end{array}\right]$$

In this study, we designate 0.40 as the threshold and in actual production the early warning threshold may be much higher than 0.40. Specifically, for each future 10 min, it corresponds a prediction, hence the data set can be further divided into six different subsets. We provide the statistics about the class imbalance rate (the number of instances belonging to the majority class divided by that of the minority class) on the data sets D1–D6 (see Figure 4). From Figure 4, it can be seen that although the data sets hold different class imbalance rates, the imbalance distribution is obvious, which might bring challenges for further modeling the classifier.

## 4. Methodology

### 4.1. Gaussian Naive Bayes Model and the Reason Why It Is Hurt by Imbalanced Data Distribution

In this section, we discuss why skewed data distribution hurts classification models based on the Bayes theorem. The reason for selecting the Bayes theorem to provide explanation is because it is the theoretical basis of statistical machine learning [32,33], and our proposed PDM inherits from it.

As we know, the original Naive Bayes model can be only used to deal with the data with discrete attributes. If the attribute space of a classification task is continuous, it needs to adopt a variant of Naive Bayes named Gaussian Naive Bayes [30,31] to model the classifier. Specifically, GNB assumes that the conditional probability of data features satisfies the Gaussian distribution, thus for any instance (*x_i_*), its conditional probability can be calculated as below:(4)$$P(x_{i}|y)=\frac{1}{\sqrt{2\pi \sigma_{y}^{2}}}e^{-\frac{{\left(x_{i}-\mu_{y}\right)}^{2}}{2\sigma_{y}^{2}}}$$
where $\mu_{y}$ and $\sigma_{y}^{2}$ denote mean and variance, respectively. These two values can be directly estimated from the original data. Here, *P*(*x_i_*|*y*) represents the conditional probability, i.e., in class *y*, the probability density of the instance *x_i_*. Then, if we know the number of instances in class *y* and the number of all instances, we can calculate the prior probability *P*(*y*). Further, based on the Bayes formula, the posterior probability *P*(*y*|*x_i_*) can be calculated by:(5)$$P(y|x_{i})=\frac{P(x_{i}|y)P(y)}{P(x_{i}|y)P(y)+P(x_{i}|\sim y)P(\text{\textasciitilde{}}y)}$$
where *P*(*y*|*x_i_*) denotes the probability of *x_i_* belonging to the class *y*.

Without loss of generality, we suppose the classification task is binary. Let $\Phi =\{(x_{i},y_{i})|x_{i}\in {ℜ}^{m},y_{i}\in \{Y_{+},Y_{-}\},1\leq i\leq q\}$ be the training data set, where $Y_{+}$ and $Y_{-}$ denote the minority and majority class, respectively. Then Φ can be divided into two different groups $\Phi_{+}=\{(x_{i},y_{i})|x_{i}\in {ℜ}^{m},y_{i}=Y_{+},1\leq i\leq q_{+}\}$ and $\Phi_{-}=\{(x_{i},y_{i})|x_{i}\in {ℜ}^{m},y_{i}=Y_{-},1\leq i\leq q_{-}\}$, where $q=q_{+}+q_{-}$ and meanwhile $q_{-}>q_{+}$. To further simplify the procedure of analysis, we suppose the number of attributes *m* = 1, and in this attribute, both classes satisfy the Gaussian distribution (see Figure 5). Then the prior probability $P(Y_{+})$, $P(Y_{-})$ and the conditional probability $P(X|Y_{+})$, $P(X|Y_{-})$ can be directly acquired. According to Bayesian formula, the posterior probability of two classes can be calculated as follows:(6)$$P(Y_{+}|X)=\frac{P(X|Y_{+})P(Y_{+})}{P(X)},\;P(Y_{-}|X)=\frac{P(X|Y_{-})P(Y_{-})}{P(X)}$$
where
(7)$$P(X)=P(X|Y_{+})P(Y_{+})+P(X|Y_{-})P(Y_{-})$$

When $P(Y_{+}|X)=P(Y_{-}|X)$, i.e., $P(X|Y_{+})P(Y_{+})=P(X|Y_{-})P(Y_{-})$, selecting the corresponding X as the separating point of two classes can guarantee the training error rate is minimal.

As shown in Figure 5, when both classes hold the same density distributions, i.e., $P(Y_{+})=P(Y_{-})$, the misclassification risk can be averagely born by both classes. While if $P(Y_{-})>P(Y_{+})$, to guarantee $P(X|Y_{+})P(Y_{+})=P(X|Y_{-})P(Y_{-})$, the separating point moves towards the minority class, which means the minority class would sacrifice more classification accuracy than that in the majority class. Specifically, the larger the difference between $P(Y_{+})$ and $P(Y_{-})$ is, the more accuracy loss the minority class would bear.

This explains the reason why skewed data distribution hurts the performance of GNB. It is only related with the prior probability, but not conditional probability density. Therefore, we can find:(8)$$P(Y_{+}|X)\propto P(X|Y_{+}),P(Y_{-}|X)\propto P(X|Y_{-})$$
which means if we can estimate the conditional probability density for each instance within each class it belongs, it can directly classify each instance. However, GNB assumes the instances belonging to each class satisfy the Gaussian distribution, which may severely misesteem the conditional probability density, further causing low modeling quality.

### 4.2. KNN-PDE-alike Probability Density Estimation Algorithm

To address the problem mentioned above, we benefit from the idea of the K-nearest-neighbors probability density estimation (KNN-PDE) strategy [34,35] to propose a robust and universal algorithm named KNN-PDE-alike.

Suppose there are *q* instances, then for each instance *x_i_*, we can find its *K*th nearest neighbors in the remainder *q* − 1 instances, and represent their distance as $d_{i}^{K}$, where *K* < *q*. It is clear that a smaller $d_{i}^{K}$ denotes a higher probability density for the instance *x_i_*, and vice versa. Considering the value principle is counterintuitive, we transform each distance to be its reciprocal, i.e., $1/d_{i}^{K}$. Next, each instances’ conditional probability density can be calculated by:(9)P(xi)=1/diKZ
where *Z* is a normalized factor that is calculated as below:(10)$$Z=\sum_{i=1}^{q}1/d_{i}^{K}$$

Actually, *P*(*x_i_*) is not the real conditional probability density, but its value reflects the same proportional relation as the conditional probability density, thus we call it relative conditional probability density. An instance holding a larger *P*(*x_i_*) value means it has a larger conditional probability density and lies in a denser region, and vice versa. In addition, from Equations (9) and (10), it is also not difficult to observe that the sum of all *q* instances’ relative conditional probability densities equals 1.

The procedure of the KNN-PDE-alike conditional probability density estimation Algorithm 1 is described as follows.
**Algorithm 1.** KNN-PDE-alike algorithm**Input**: a data set Φ = {*x_i_* | *x_i_*∈**R***^m^*, 1 ≤ *i* ≤ *q*}, the neighborhood parameter *K.***Output**: a 1 × *q* vector *CPD* to record the relative conditional probability density of all instances.**Procedure**:**1.** For each instance *x_i_*, find its *K*th nearest neighbor, and record their distance $d_{i}^{K}$;**2.** Calculate the normalized factor *Z* by Equation (10);**3.** Calculate the relative conditional probability density *P*(*x_i_*) by Equation (9);**4.** Record the relative conditional probability density one by one into *CPD* and output it.

### 4.3. Probability Density Machine

Based on KNN-PDE-alike algorithm, we can precisely estimate the relative conditional probability density of each instance regardless of data distribution types. Then, an unbiased prediction can be provided. The proposed unbiased CIL prediction algorithm is called the probability density machine (PDM). Specifically, considering in imbalanced data, different classes hold a different number of instances, which means the relative conditional probability density of different classes are scaled in different scales, hence it is difficult to directly compare the relative conditional probability density of two instances belonging to two different classes. To deal with this problem, a pre-normalization process should be firstly conducted to unify the dimension of the relative conditional probability density from different classes. Suppose the instance *x_i_* is from the majority class, then its normalized conditional probability density can be calculated as below:(11)P(xi)=1/diKZ−×CIR
where *Z*_−_ and *CIR* denote the normalized factor of majority class and class imbalance ratio, respectively.

The other parameter which may influence the estimating accuracy of the relative conditional probability density is the neighborhood factor *K*. Obviously, it is inappropriate to designate a uniform value for the parameter *K* as there exists a significant difference about the number of instances belonging to different classes. In addition, we note it is also inappropriate to assign an oversize or too small value for the parameter *K*. If the *K* value is too large, the distinction of the relative conditional probability density from different instances would become ambiguous, but if the *K* value is too small, it would reflect more about a narrow local probability density distribution, but not the global probability density distribution. In this work, we suggest designating *K* as $\sqrt{q}$ by default.

The procedure of the PDM Algorithm 2 is described as follows.
**Algorithm 2**. PDM algorithm**Input**: an imbalanced training set Φ = {(*x_i_*, *y_i_*) | *x_i_*∈**R***^m^*, 1 ≤ *I* ≤ *q*, *y_i_*∈{*Y*_+_, *Y*_−_ }}, a test instance *x*’.**Output**: a predictive class label for *x*’.**Training Procedure**:**1.** For majority class *Y*_−_ and minority class *Y*_+_, extract their corresponding instances into Φ_−_ and Φ_+_ from Φ, count their number of instances and record them as *q*_−_ and *q*_+_, respectively;**2.** Calculate *CIL* which equals |Φ_−_|/|Φ_+_|;**3.** For two classes *Y*_−_ and *Y*_+_, set the corresponding parameter *K* as $\sqrt{q_{-}}$ and $\sqrt{q_{+}}$, respectively;**4.** For two classes, call KNN-PDE-alike algorithm to obtain and record the corresponding normalized factors *Z*_−_ and *Z*_+_.**Testing Procedure**:**1.** For class *Y*_−_, put *x*’ and the corresponding Φ_−_, *K*_−_ and *Z*_−_ into KNN-PDE-alike algorithm to obtain the corresponding relative conditional probability density *P*_−_(*x*’);**2.** For class *Y*_+_, put *x*’ and the corresponding Φ_+_, *K*_+_ and *Z*_+_ into KNN-PDE-alike algorithm to obtain the corresponding relative conditional probability density *P*_+_(*x*’);**3.** For class *Y*_−_, call *CIL* to adjust its relative conditional probability and obtain the normalized probability density *P*_−_(*x*’) by Equation (11);**4.** Compare *P*_−_(*x*’) and *P*_+_(*x*’), if *P*_−_(*x*’)> *P*_+_(*x*’), predicting *x*’ into *Y*_−_, otherwise, *x*’ is predicted into *Y*_+_;**5.** Output the class label of the test instance *x*’.

## 5. Experiments

### 5.1. Experimental Settings

In the experiment, we designated *l* = 24, *t* = 6, $K=\sqrt{q}$ by default, which were indicated in the sections above. Specifically, due to *t* = 6, we need to train six different classification models on each data set as our proposed PDM model or other classification models can only deal with single-output problem. In addition, we compared the proposed PDM algorithm with GNB [30,31] and its combination with several traditional sampling strategies, including RUS [22], ROS [22] and SMOTE [23]. We also compared it with several state-of-the-art class imbalance learning algorithms, including GL [24], FSVM-CIL [26] and ODOC-ELM [28]. All comparison algorithms used the default parameter settings in the corresponding references.

As for the performance evaluation metric, we all know accuracy is not an excellent metric to evaluate the quality of class imbalance learning models, thus we adopted a popular metric called F-measure in this paper. F-measure can be calculated as below:(12)$$\mathrm{F-measure}=\frac{2\times \mathrm{Precision}\times \mathrm{Recall}}{\mathrm{Precision}+\mathrm{Recall}}$$

In fact, F-measure represents a tradeoff between precision and recall.

Finally, in order to impartially compare the performance of various algorithms, we adopted the randomly external 5-fold cross validation 10 times to calculate the average performance as the final results.

### 5.2. Results and Discussions

Table 2 shows the experimental results of 8 compared algorithms. From the results, we can safely draw several conclusions as follows:(1)On gas concentration data, two traditional oversampling strategies, i.e., ROS and SMOTE, seem to be impossible for promoting the quality of the classification model, while RUS sometimes presents a little better performance than GNB. We believe it associates with the structural and distribution complexity of gas concentration monitoring data. On this kind of data, ROS makes the model extremely overfitting, and SMOTE generates many synthetic instances on inappropriate positions. GL-GNB, which considers the distribution during sampling, alleviates the problem of ROS and SMOTE to some extent. However, the performance promotion by adopting GL-GNB seems to be restricted as on D4 and D5, it produces worse performance than GNB.(2)Class imbalance rate can influence the performance of various algorithms to some extent, including the proposed PDM algorithm. It can clearly be observed that the worse F-measure values exist on those two highly imbalanced data sets, namely D3 and D6, while on other data sets, the classification performance is obviously better. We believe it is related to a rare number of minority training instances, which are not enough to precisely reconstruct the probability distribution of the minority class.(3)On most data sets, two algorithm-level algorithms, i.e., FSVM-CIL and ODOC-ELM, perform significantly better than those sampling-based strategies. Of course, they are both more sophisticated than those sampling algorithms. FSVM-CIL needs to explore data distribution and assign individual cost weight for each instance, while ODOC-ELM needs to adopt the random optimization algorithm to iteratively search the best threshold. We also note that on highly imbalanced data sets, e.g., D3 and D6, FSVM-CIL outperforms ODOC-ELM, while on the other data sets, ODOC-ELM performs better.(4)The proposed PDM algorithm outperforms all other solutions. In fact, it produced the best result on the most predicted time points of each data set. Specifically, in comparison with the GNB, the performance of PDM was improved 13.1~24.7%, while compared with several other algorithms, the performance of PDM promotes 0.4~40.7%. The results verifies the effectiveness and superiority of the proposed PDM algorithm.

Figure 6 shows the multi-step prediction results of the PDM algorithm on each data set. In Figure 6, it is not difficult to observe that the predicted quality always declines with the antedisplacement of the predicting time. It is consistent with our intuition as the experience should have a stronger association with a nearer future status. Therefore, we think the early warning model can only implement short-time prediction.

### 5.3. Significance Statistical Analysis

Next, we tested the actual difference between the PDM and the other compared algorithms in statistics. Specifically, the critical difference (CD) metric is used to show the difference of various algorithms. Figure 7 shows the CD diagram at a standard level of significance *α* = 0.05, where the average ranking of each algorithm is marked along the axis (higher rankings to the left). In the CD diagram, if a group of algorithms are not significantly different under the Nemenyi test [36,37], these algorithms are connected by a thick line.

From the results shown in Figure 7, we observe that the PDM algorithm achieves the statistically superior performance over all other algorithms except ODOC-ELM, and although we cannot say it has significant difference with ODOC-ELM algorithm, it has a lower average rank than ODOC-ELM. To summarize, the proposed PDM algorithm is a better choice than the compared algorithms. 

### 5.4. Discussion about the Parameters

To accurately estimate the relative conditional density in PDM, the choice of parameter *K* is very important. As indicated in Section 3, it is inappropriate to assign an oversize or too small value for the parameter *K*. If the *K* value is too large, the distinction of relative conditional probability density from different instances would become ambiguous, but if the *K* value is too small, it would reflect more about a narrow local probability density distribution, but not the global probability density distribution. To verify the deduction, we varied the parameter *K* in the range of $\left\{\sqrt{q}/4,\sqrt{q}/2,\sqrt{q},2\sqrt{q},4\sqrt{q}\right\}$. The variance of the F-measure performance with the variance of the parameter *K* is presented in Table 3. It is clear that when *K* is assigned as a value between $\sqrt{q}/2$ and $2\sqrt{q}$, the performance of PDM can be safely guaranteed.

Next, we also focused on the impact of another parameter *l* which defines how long of recent experiences are useful for predicting the future status, and determines the dimension of attribute space. We varied *l* in the range of {6, 12, 24, 48, 72}, i.e., taking advantage of the experience of recent 1, 2, 4, 8 and 12 h, to observe the performance variance of the PDM. The performance variance with the variance of the parameter *l* is presented in Table 4. The results in Table 4 reflect that adopting the experience of recent 2~4 h is most appropriate for predicting the near future gas concentration status. An oversize *l* may insert some irrelevant noises, while a too small *l* may lack some significant information.

## 6. Conclusions

In this study, we focused on the early warning issue of gas concentration in coal mine production. Specifically, we consider it as an imbalanced binary-class classification issue, and in the context of the Naive Bayes theory, we propose a novel class imbalance learning algorithm called the probability density machine. By six real gas concentration monitoring data sets acquired from a coal mine in Datong city, Shanxi Province, China, the effectiveness and superiority of the proposed PDM algorithm was verified.

The contributions of this study can be concluded as follows:(1)We consider the early warning issue of gas concentration in coal mine production as a classification issue, and note its characteristics of class imbalance and sophisticated distribution;(2)In context of the Naive Bayes theory, we analyzed why imbalanced data distribution can hurt predicted models in theory;(3)A novel class imbalance learning algorithm called the probability density machine was used to promote the accuracy of early warning of gas concentration in coal mine production.

In future work, except the gas concentration data, other synchronously occurring sensing or monitoring data will also be considered to be added into the predicting model for the purpose of improving the predicting accuracy.

## Figures and Tables

**Figure 1 sensors-21-05730-f001:**
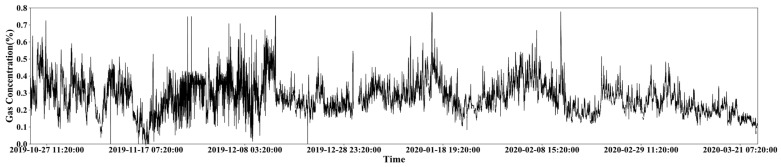
An example of gas concentration variation trend over 5 months at a monitoring site.

**Figure 2 sensors-21-05730-f002:**
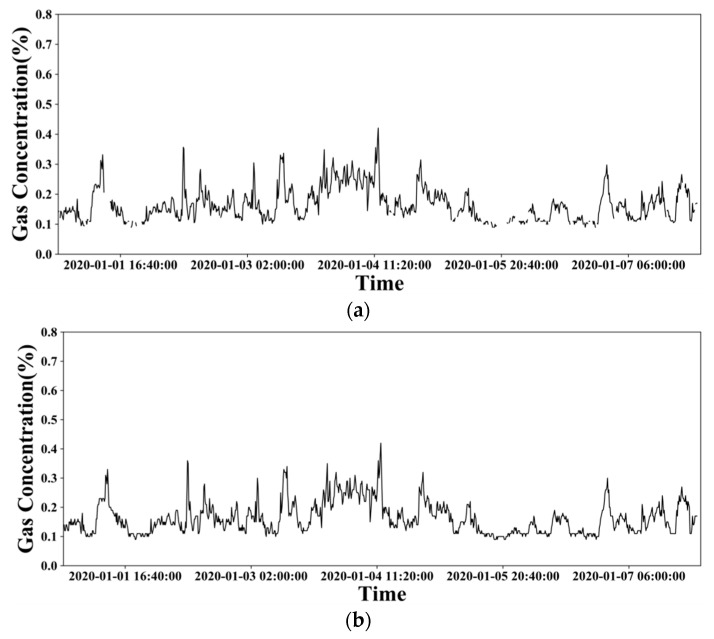
The gas concentration variation trend of a monitoring point within a week before (**a**) and after (**b**) missing data imputation.

**Figure 3 sensors-21-05730-f003:**
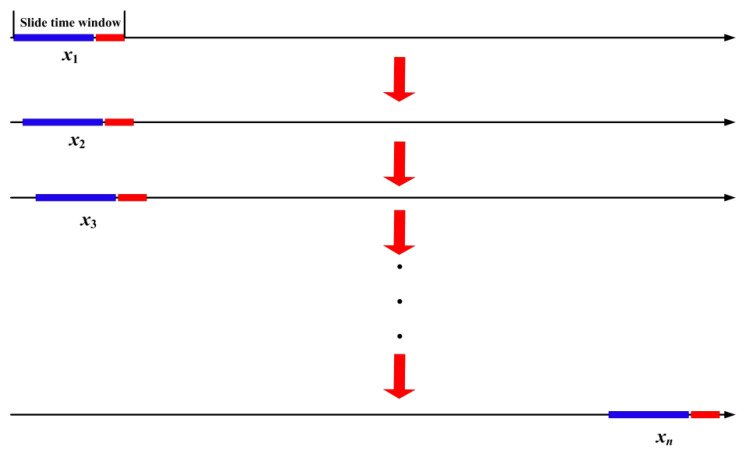
A graphical representation to describe how to generate instances, where the blue bar and red bar in the slide time window indicate the attributes and the corresponding predicting values, respectively.

**Figure 4 sensors-21-05730-f004:**
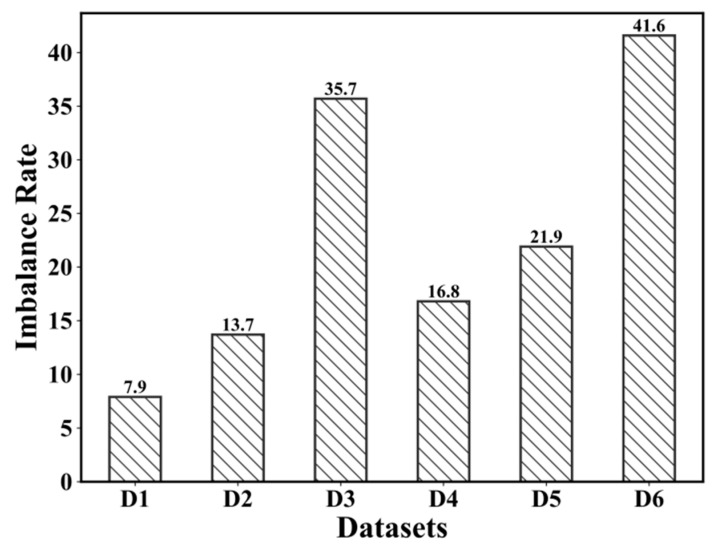
Class imbalance rate on each data set.

**Figure 5 sensors-21-05730-f005:**
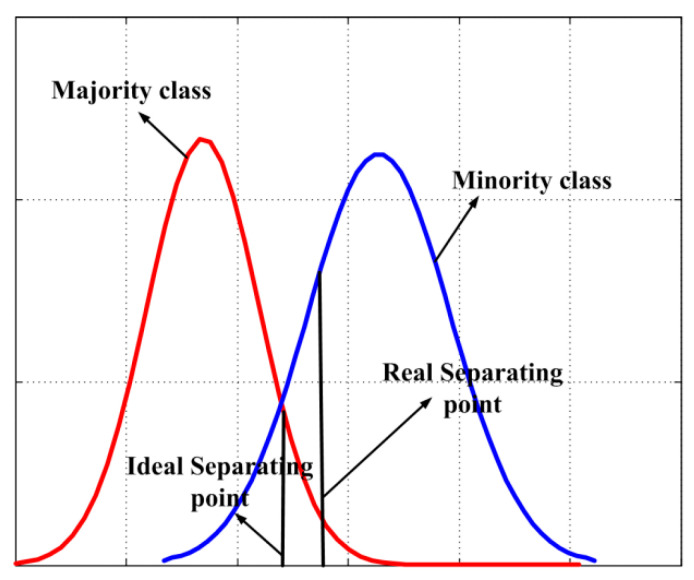
An example that explains why imbalanced data distribution can hurt classification models.

**Figure 6 sensors-21-05730-f006:**
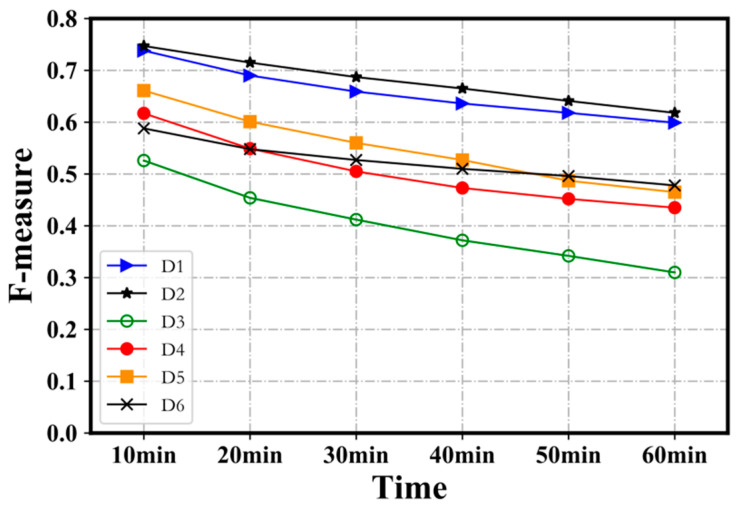
Performance comparison of the PDM algorithm on multiple future time points.

**Figure 7 sensors-21-05730-f007:**
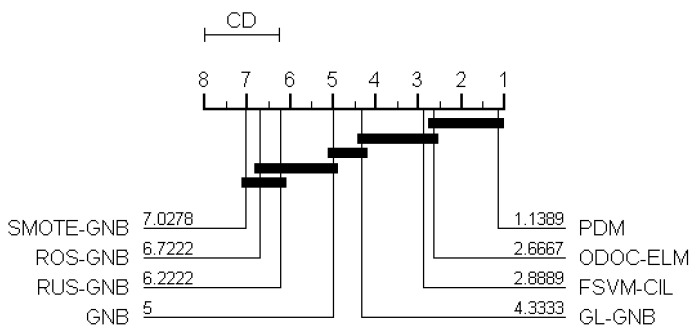
CD diagram.

**Table 1 sensors-21-05730-t001:** Statistics of gas concentration data at each monitoring point.

Data Set	Total	Loss	Loss Rate
D1	22,032	1354	6.1%
D2	22,032	1845	8.4%
D3	22,032	2237	10.2%
D4	22,032	2532	11.5%
D5	22,032	3155	14.3%
D6	22,032	3423	15.5%

**Table 2 sensors-21-05730-t002:** F-Measure of future one-hour early warning results of each algorithm on six data sets, where the best result is highlighted in bold.

Data Set	Algorithm	10 min	20 min	30 min	40 min	50 min	60 min	Average
D1	PDM	**0.738**	**0.690**	**0.659**	**0.636**	**0.618**	**0.599**	**0.657**
	GNB	0.560	0.537	0.522	0.511	0.503	0.494	0.521
	RUS-GNB	0.601	0.569	0.548	0.534	0.524	0.511	0.548
	ROS-GNB	0.550	0.529	0.513	0.502	0.492	0.484	0.512
	SMOTE-GNB	0.549	0.527	0.512	0.499	0.489	0.481	0.510
	GL-GNB	0.629	0.581	0.563	0.549	0.537	0.531	0.565
	FSVM-CIL	0.633	0.597	0.601	0.585	0.576	0.544	0.589
	ODOC-ELM	0.675	0.630	0.599	0.572	0.596	0.563	0.606
D2	PDM	**0.747**	**0.715**	**0.687**	**0.665**	**0.641**	**0.618**	**0.679**
	GNB	0.520	0.497	0.481	0.468	0.453	0.441	0.477
	RUS-GNB	0.434	0.417	0.403	0.391	0.381	0.372	0.400
	ROS-GNB	0.503	0.480	0.464	0.448	0.436	0.427	0.460
	SMOTE-GNB	0.507	0.483	0.465	0.448	0.436	0.425	0.461
	GL-GNB	0.528	0.525	0.510	0.533	0.479	0.456	0.505
	FSVM-CIL	0.499	0.486	0.472	0.481	0.466	0.452	0.476
	ODOC-ELM	0.691	0.652	0.639	0.601	0.598	0.573	0.626
D3	PDM	**0.526**	**0.454**	**0.412**	**0.372**	**0.342**	**0.310**	**0.403**
	GNB	0.237	0.217	0.206	0.197	0.191	0.183	0.205
	RUS-GNB	0.368	0.287	0.225	0.173	0.138	0.108	0.217
	ROS-GNB	0.222	0.203	0.192	0.182	0.174	0.167	0.190
	SMOTE-GNB	0.221	0.201	0.186	0.178	0.172	0.164	0.187
	GL-GNB	0.307	0.251	0.204	0.187	0.192	0.175	0.219
	FSVM-CIL	0.519	0.432	0.377	0.291	0.286	0.259	0.361
	ODOC-ELM	0.428	0.295	0.356	0.272	0.261	0.253	0.311
D4	PDM	**0.617**	**0.549**	**0.505**	0.473	**0.452**	0.435	**0.505**
	GNB	0.416	0.389	0.372	0.363	0.355	0.349	0.374
	RUS-GNB	0.215	0.209	0.204	0.201	0.198	0.196	0.204
	ROS-GNB	0.398	0.373	0.358	0.346	0.337	0.331	0.357
	SMOTE-GNB	0.399	0.370	0.354	0.342	0.333	0.327	0.354
	GL-GNB	0.419	0.381	0.374	0.350	0.353	0.332	0.368
	FSVM-CIL	0.514	0.453	0.446	0.429	0.398	0.401	0.440
	ODOC-ELM	0.555	0.507	0.481	**0.486**	0.440	**0.439**	0.485
D5	PDM	**0.661**	**0.601**	**0.560**	**0.527**	**0.487**	**0.465**	**0.550**
	GNB	0.360	0.330	0.307	0.292	0.276	0.263	0.305
	RUS-GNB	0.575	0.519	0.482	0.455	0.418	0.352	0.467
	ROS-GNB	0.333	0.306	0.286	0.271	0.258	0.247	0.284
	SMOTE-GNB	0.333	0.304	0.286	0.268	0.256	0.243	0.282
	GL-GNB	0.349	0.321	0.304	0.288	0.265	0.258	0.298
	FSVM-CIL	0.507	0.486	0.454	0.429	0.401	0.382	0.443
	ODOC-ELM	0.492	0.471	0.460	0.443	0.399	0.385	0.442
D6	PDM	**0.588**	0.548	**0.527**	**0.510**	0.496	0.478	**0.525**
	GNB	0.320	0.294	0.277	0.266	0.258	0.251	0.278
	RUS-GNB	0.139	0.130	0.125	0.120	0.116	0.113	0.124
	ROS-GNB	0.305	0.276	0.262	0.250	0.241	0.232	0.261
	SMOTE-GNB	0.321	0.284	0.267	0.255	0.244	0.235	0.268
	GL-GNB	0.356	0.332	0.309	0.311	0.289	0.273	0.312
	FSVM-CIL	0.576	**0.553**	0.515	0.508	**0.502**	**0.484**	0.523
	ODOC-ELM	0.389	0.304	0.327	0.299	0.271	0.254	0.307

**Table 3 sensors-21-05730-t003:** Variance of the F-measure performance with the variance of the parameter *K*.

**Data Set**	$\sqrt{q}/4$	$\sqrt{q}/2$	$\sqrt{q}\;$	$2\sqrt{q}$	$4\sqrt{q}$
D1	0.665	**0.66** **8**	0.657	0.648	0.636
D2	0.683	**0.685**	0.679	0.669	0.657
D3	0.376	0.398	0.403	**0.415**	0.401
D4	0.479	0.497	**0.505**	0.502	0.495
D5	0.536	**0.551**	0.550	0.536	0.503
D6	0.501	0.524	**0.525**	0.516	0.503

**Table 4 sensors-21-05730-t004:** Variance of the F-measure performance with the variance of the parameter *l*.

Data Set	6	12	24	48	72
D1	0.642	**0.6** **84**	0.657	0.639	0.614
D2	0.659	0.671	**0.679**	0.664	0.663
D3	0.395	**0.417**	0.403	0.357	0.344
D4	0.488	**0.538**	0.505	0.470	0.444
D5	0.531	0.547	**0.550**	0.521	0.515
D6	0.511	0.520	**0.525**	0.506	0.488

## Data Availability

The data are collected from a coal mine in Datong city, Shanxi Province, China, and are not being released due to privacy issues. If necessary, please contact the author.

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
