# Peer review of "Early Warning of Gas Concentration in Coal Mines Production Based on Probability Density Machine"

_sensors, 2021, doi:10.3390/s21175730_

Round 1

Reviewer 1 Report

A new approach is proposed for gas concentration estimation in coal mines production, addressing simultaneously the inherent problem of imbalanced data, which shown a superior performance when compared to current methods.

The work is well organized and structured. A simple but comprehensive written has been used and the methodology implemented clearly described. Moreover, a very complete literature review is present, supported on updated references, underling the actual limitation of existing methods and showing how the proposed approach will overcome it.

Different strategies were adopted to deal with the high data missing rate, by selecting only six sensor with less than 20% missing rate and the application of an imputation method. A new strategy has been proposed to address the problem of imbalanced data. In summary, a very complete approach was adopted for data preparation aiming to assure data quality. In addition,  a 10 runs under a 5-fold cross validation approach was applied, which seems to be a an adequate option for the type and amount of data.

There is a specific question that needs to be clarified, which relates with the different performance achieved by the PDM algorithm on the six datasets. Particularly, a considerable difference is observed on D2 and D3 performance. What are the main reasons for such a considerable difference? A comment should be included in the paper.

In conclusion, the proposed approach shows a superior performance, which represent a considerable contribution towards a more efficient prediction of gas concentration in coal mines production.

In the light of that, the work is proposed for publication in Sensors journal.

Author Response

For Reviewer 1:

Q1: There is a specific question that needs to be clarified, which relates with the different performance achieved by the PDM algorithm on the six datasets. Particularly, a considerable difference is observed on D2 and D3 performance. What are the main reasons for such a considerable difference? A comment should be included in the paper.

Answers: Thanks for the valuable suggestions from the reviewer. In fact, we discussed it when we analyze the results in Table 2. D3 and D6 have significantly higher class imbalance rates than the other data sets, causing it is more difficult to construct excellent models on these two data sets. In the revised manuscript, we revise the description to highlight it as follows: Class imbalance rate could influence the performance of various algorithms to some extent, including the proposed PDM algorithm. It can clearly observe that the worse F-measure values exist on those two highly imbalanced data sets, namely D3 and D6, while on other data sets, the classification performance is obviously better. We believe it is related with rare number of minority training instances, which are not enough to precisely reconstruct the probability distribution of the minority class.

Reviewer 2 Report

This paper considered the early warning of gas concentration as a binary-class problem and  proposed Probability Density Machine (PDM) algorithm to deal with the imbalance problem.

Suggestions:

  • The works that consider gas early warning prediction as a binary classification problem should be investigated.
  • The GNB algorithm is indeed senstive to the class imbalance, and there are many other algorithms like SVM, DT, MLP and so on. The motivation of choosing the GNB as baseline should be described.
  • How to deal with the multiple outputs based on the proposed PDM algorithm should be presented.  
  • The results show that the proposed PDM algorithm is superior to other compared algorithms. However, the compared algorithms are all proposed for  a long time. Some recently proposed algorithms should be compared. 

Author Response

For Reviewer 2:

Q1: The works that consider gas early warning prediction as a binary classification problem should be investigated.

Answers: Yes, we have investigated its rationality after discussed with the safe production engineers in coal mines. In practical coal mining production, it is not necessary to accurately predict gas concentration values over a period of future time (regression), but judge whether it has a high risk to impact safety production or not is enough (classification). We have added the exploration in Section 1. Introduction as follows: It is reasonable that regarding gas early warning prediction as a binary classification issue, as in practical coal mining production, it is not necessary to accurately predict gas concentration over a period of future time, but judge whether it has a high risk to impact safety production or not is enough.

Q2: The GNB algorithm is indeed senstive to the class imbalance, and there are many other algorithms like SVM, DT, MLP and so on. The motivation of choosing the GNB as baseline should be described.

Answer: Yes, we know nearly all classification models are sensitive to imbalanced data distribution as most of them are constructed based on the empirical risk minimization theory, even SVM which is based on structural risk minimization is still sensitive to skewed data distribution. There are two reasons for us to select GNB: 1. GNB is based on Bayes theory that has a strong theoretical basis of statistical machine learning; 2. our proposed PDM is based on GNB but not the other classification algorithms. That’s why we select GNB as the baseline. We added the reasons in Section 1 and Section 4 in the revised manuscript.

Q3: How to deal with the multiple outputs based on the proposed PDM algorithm should be presented.

Answer: Thanks for the suggestions from the reviewer. The PDM algorithm can be only used to deal with two-class and multiclass classification problems, but it is helpless to address multiple outputs problem directly. In our experiments, six PDM is respectively trained to predict future one hour gas concentrations, where each PDM corresponds a ten minutes predicted reader. To avoid the readers to confuse, we have added the explanation in the experimental setting section as flows: Specifically, due to t=6, we need to train 6 different classification models on each data set as no matter our proposed PDM model or other classification models can’t only deal with single-output problem.

Q4: The results show that the proposed PDM algorithm is superior to other compared algorithms. However, the compared algorithms are all proposed for a long time. Some recently proposed algorithms should be compared.

Answer: We wish to show our specifical gratitude for the suggestions from the reviewer. Yes, although RUS, ROS and SMOTE are most popular and widely used class imbalance learning techniques in practical applications, they are too old, thus influence the persuasion of experimental comparisons. In the revised paper, we added the description about class imbalance learning techniques, and added the experimental comparisons with three state-of-the-art and representative class imbalance learning algorithms, including a sampling algorithm, a cost-sensitive learning algorithm and a threshold-moving algorithm. Based on the new experimental results, we have updated our discussion in the revised manuscript.

Round 2

Reviewer 2 Report

It is reasonable that regarding gas early warning prediction as a binary classification issue. It will be better if some previous works that considered the gas early warning preduiction as a binary classification problem is introduced.

Author Response

Thanks for the suggestions from the reviewer. We have tried our best to search papers which consider using classification models to predict gas early warning. There are two work satisfy this requirement to some extent. One uses multiple classification models with optimization approach to provide a relatively accurate warning for future 3 minutes’ methane emission, and the other one develops a gas outburst early warning system by adopting an entropy-weight Bayes inference model. They are both not totally consistent with our applications, as the former constructs methane emission warning model but not gas early warning prediction, and meanwhile it could only predict future 3 minutes’ status, which is not practical in real-world application, while the latter considers multiple factors but not only gas concentration. Meanwhile, they both ignore the influence of imbalanced data distribution existing in gas concentration data. However, both of these two studies indicate that it is reasonable for regarding gas early warning prediction as a binary classification issue. In the revised manuscript, we have added the introduction about these two studies in Introduction section. We expect it can answer the confusion from the reviewer and future potential readers well.

Specifically, we have highlighted all revisions in red in new manuscript. We also revised some spelling and grammar errors throughout the manuscript.